# Baseline Plasma Tumor DNA (ctDNA) Correlates with PSA Kinetics in Metastatic Castration-Resistant Prostate Cancer (mCRPC) Treated with Abiraterone or Enzalutamide

**DOI:** 10.3390/cancers14092219

**Published:** 2022-04-29

**Authors:** Vincenza Conteduca, Chiara Casadei, Emanuela Scarpi, Nicole Brighi, Giuseppe Schepisi, Cristian Lolli, Giorgia Gurioli, Ilaria Toma, Giulia Poti, Alberto Farolfi, Ugo De Giorgi

**Affiliations:** 1IRCCS Istituto Romagnolo Per Lo Studio Dei Tumori (IRST) “Dino Amadori”, via Piero Maroncelli 40, 47014 Meldola, Italy; chiara.casadei@irst.emr.it (C.C.); emanuela.scarpi@irst.emr.it (E.S.); nicole.brighi@irst.emr.it (N.B.); giuseppe.schepisi@irst.emr.it (G.S.); cristian.lolli@irst.emr.it (C.L.); giorgia.gurioli@irst.emr.it (G.G.); alberto.farolfi@irst.emr.it (A.F.); ugo.degiorgi@irst.emr.it (U.D.G.); 2Department of Medical and Surgical Sciences, Unit of Medical Oncology and Biomolecular Therapy, University of Foggia, Policlinico Riuniti, 71122 Foggia, Italy; 3Department of Medical Oncology, Card. G. Panico Hospital of Tricase, 73039 Tricase, Italy; ilaria.toma88@gmail.com; 4Istituto Dermopatico dell’Immacolata, IDI-IRCCS, 00167 Rome, Italy; giuliapoti@gmail.com

**Keywords:** metastatic castration-resistant prostate cancer, AR copy number, circulating tumor DNA, biomarkers

## Abstract

**Simple Summary:**

Prostate cancer is a very common disease in men. Nowadays several life-prolonging therapies are available, also efficient in the metastatic setting. However, it is important to choose the best approach for each patient, and in this context molecular biomarkers are fundamental. Baseline high circulating tumor DNA (ctDNA) fraction in plasma and androgen receptor (AR) copy number (CN) gain correlates with worse outcomes. This study investigates correlation between PSA response endpoints, plasma DNA analysis and progression free/overall survival, underling the importance of a multimodal approach to early predict outcome.

**Abstract:**

Background: Baseline high circulating tumor DNA (ctDNA) fraction in plasma and androgen receptor (AR) copy number (CN) gain identify mCRPC patients with worse outcomes. This study aimed to assess if ctDNA associates with PSA kinetics. Methods: In this prospective biomarker study, we evaluate ctDNA fraction and AR CN from plasma samples. We divided patients into high and low ctDNA level and in AR gain and AR normal. Results: 220 baseline samples were collected from mCRPC treated with abiraterone (n = 140) or enzalutamide (n = 80). A lower rate of PSA decline ≥ 50% was observed in patients with high ctDNA (*p* = 0.017) and AR gain (*p* = 0.0003). Combining ctDNA fraction and AR CN, we found a different median PSA progression-free survival (PFS) among four groups: (1) low ctDNA/AR normal, (2) high ctDNA/AR normal, (3) low ctDNA/AR gain, and (4) high ctDNA/AR gain (11.4 vs. 5.0 vs. 4.8 vs. 3.7 months, *p* < 0.0001). In a multivariable analysis, high ctDNA, AR gain, PSA DT, PSA DT velocity remained independent predictors of PSA PFS. Conclusions: Elevated ctDNA levels and AR gain are negatively and independently correlated with PSA kinetics in mCRPC men treated with abiraterone or enzalutamide.

## 1. Introduction

Prostate cancer is one of the most common tumor in men and the second leading cause of cancer-related death worldwide [1,2]. In prostate cancer, androgen receptor (AR) has a central role in promoting the progression of prostate cancer and the inhibition of AR signaling by using androgen deprivation therapy (ADT) that represents the first treatment for castration-sensitive prostate cancer [3]. Hormone-naive, or ‘castrate sensitive’ advanced prostate cancer (CSPC) is characterized by non-castrate testosterone levels and includes clinical disease states ranging from patients with prostate specific antigen (PSA) recurrence to those patients with metastatic prostate cancer detected by imaging. Regardless of stage, nearly all patients treated with ADT with or without additional systemic therapy will eventually develop castration-resistant prostate cancer (CRPC), defined as radiographic progression and/or rising PSA despite castrate levels of testosterone. [4]. In recent years, several drugs have been largely approved in mCRPC, such as docetaxel [5], cabazitaxel [6] and Radium-223 [7], but above all second generation AR signaling inhibitors (ARSI) such as abiraterone [8,9,10] and enzalutamide [11,12]. In this context, it is important to select the best therapeutic option for each patient. Currently, it was generally believed that CRPC is “hormone refractory” based on the notion that AR signaling was dispensable to the biology of this disease state. However, early genomic studies indicated that 30% of CRPC patients harbour high-level amplification of the *AR* locus in late stage tumors. This spurred inquiries by numerous groups to re-address the possible contribution of residual androgens remaining after castration as well as the AR itself to the progression of CRPC and its continued growth. Collectively, these efforts revealed that genetic alterations at the AR locus act to restore AR signaling in the setting of low androgens [13,14]. In fact, most patients still had a disease dependent on AR signaling because of the acquisition of AR gene mutations, amplification or other mechanism of re-activation of AR [15,16,17,18,19] (Figure 1).

Despite an effective and often durable response, resistance to AR-directed therapies ultimately occurs [20].

More recently, it has been recognized that a subset of CRPC tumors may exhibit phenotypic plasticity and can switch to an alternative lineage that is less dependent on the canonical AR pathway as a means to evade AR signaling inhibitors (ARSI), becoming in some cases, very aggressive with histologic transformation into poorly differentiated state and atypical spread, and/or progression with low or non-rising PSA levels developed [21]. 

Historically, contradictory findings were reported about the surrogacy and predictive performance of PSA kinetics with different parameters (Figure 2) in mCRPC patients [22,23,24,25,26,27], but these results mainly derived from studies of mCRPC patients receiving chemotherapy. However, the efficacy of ARSI in terms of survival and PSA response confirms that androgen signaling remains important in mCRPC and that PSA kinetics may be largely related to activity at the AR [28,29]. In this context, the results collected in two phase III studies, COU-AA-301 and COU-AA-302, provided the evidence that survival outcomes can be adequately predicted through PSA kinetics because of its strong association with overall survival (OS) and that the Prentice criteria for surrogacy were met for these PSA kinetics endpoints [30].

Currently, standard disease evaluation recommended by the Prostate Cancer Clinical Trials Working Group 3 (PCWG3) guidelines [31] provide imaging tests and PSA assessments. In addition, PCWG3 criteria include blood-based diagnostics, such as circulating nucleic acids, to better characterize disease biology and identify potential predictive molecular biomarkers.

Prostate cancer is a heterogeneous disease that changes during its natural history under the pressure of different treatments. [32]. Circulating tumor DNA derived from patient plasma (ctDNA) is a promising minimally invasive biomarker, given its high concordance with matched metastatic biopsies [33,34,35]. ctDNA represents a useful tool for the understanding of tumor characteristics in real-time and could help physicians to track treatment efficacy and eventual resistance mechanisms [36].

Quantitative and qualitative alteration (such as AR copy number variation) detected in ctDNA have been proposed as significant predictors of clinical outcomes. Particularly, their role as a prognostic/predictive biomarker in mCRPC has been established by using blood samples from patients before starting ARSI [37,38]. This study aimed to assess if plasma DNA analysis and AR status associates with PSA kinetics.

## 2. Materials and Methods

This was a single-institution analysis of plasma samples collected prospectively in a study with the primary objective of biomarker evaluation (REC 2192/2013). Participants were required to have histologically-confirmed prostate adenocarcinoma without neuroendocrine differentiation, progressive disease despite “castration levels” of serum testosterone (<50 ng/dL), on-going LHRH analogue treatment or prior surgical castration. Patients received ARSI treatment with abiraterone 1g once a day and prednisone 5mg twice daily or enzalutamide 160 mg once daily as first-line or second-line therapy. ARSI were administered continuously until evidence of progression disease or unacceptable toxicity. Serum PSA was evaluated within first 3 days of therapy and monthly thereafter. Plasma ctDNA was collected only at baseline, in particular at the same time as for PSA. Radiographic disease was assessed with the use of computed tomography and bone scan at the time of screening and every 12 weeks on treatment.

Serial blood samples were collected pre-treatment (within 3 days before commencing ARSI therapy) and at different time-points (when available): on treatment and/or at progression disease (PD).

The study was conducted in accordance with the Declaration of Helsinki and the Good Clinical Practice guidelines of the International Conference of Harmonization. Written informed consent was obtained from all patients.

Blood samples were processed into plasma within 3 h of collection. Circulating DNA was extracted from 1 to 2 mL of plasma from each patient using the QIAamp Circulating Nucleic Acid Kit (Qiagen) and quantified using the high-sensitivity Quant-iT PicoGreen double-stranded DNA Assay Kit (Invitrogen) [37,38,39].

In plasma and patient-matched germline DNA, targeted next-generation sequencing (NGS) was performed on the PGM Ion Torrent using a 316 or 318 Chip to account for 1000× expected coverage per target. We estimated the global tumor content for each sequential plasma sample from study patients by using the approach previously developed [22,37], which extends the CLONET framework [23].

We assessed AR amplification in circulating plasma DNA by digital droplet polymerase chain reaction (ddPCR) using three reference genes: NSUN3, ElF2C1, and AP3B1, and ZXDB at Xp11.21 as a control gene not involving the whole arm of chromosome. Each PCR reaction was prepared with 1–2 ng DNA [39].

Primary endpoint of the study was to determine if ctDNA associates with PSA kinetics. Secondary endpoints were to evaluate the role of ctDNA and AR gain in monitoring the response to treatment with abiraterone or enzalutamide. The radiographic and biochemical response were defined according to PCWG3 criteria [31]. ctDNA high was defined as ≥0.180 according to median value. PSA kinetics was described analyzing various PSA value longitudinally collected. PSA doubling time (PSA DT) was defined as the number of months required for the PSA level to double and may be associated with prostate cancer cell proliferation [40]. The models were developed using data from patients who received at least one dose of abiraterone or enzalutamide and for whom at least one post-treatment PSA was available.

OS was calculated from the first day of therapy until death or last follow-up. Radiographic progression-free survival (PFS) was calculated from the start of each therapy to the date of progression disease or death, whichever occurs first, or last tumor evaluation.

Survival curves were estimated by the Kaplan-Meier method and were compared using the log-rank test. Cox regression models were utilized to investigate potential factors which could predict PFS and OS and to estimate hazard ratios (HRs) and their 95% confidence interval (CI). In order to obtain a parsimonious model, multivariate analyses were performed using Cox regression models including variables statistically significant at univariate analysis.

Continuous variables were summarized by descriptive statistics (number of cases, median, interquartile range-IQR) and were compared using median test. Categorical variables were reported using counts of patients and percentages and were compared using Chi-squared test.

All *p*-values were two-sided and a *p* < 0.05 was considered as statistically significant. Statistical analyses were performed with SAS 9.4 software (SAS Institute, Cary, NC, USA).

## 3. Results

### 3.1. Patient and Plasma Sample Characteristics

Between March 2011 and June 2016, 220 mCRPC patients treated with ARSI (140 abiraterone, 80 enzalutamide) at IRCCS Istituto Romagnolo Per Lo Studio Dei Tumori (IRST) “Dino Amadori” for whom biological sample were available were included. Overall, median age was 74 years [interquartile range (IQR) 69–79] and 205 patients (93.2%) had a performance status (PS) of 0–1. Gleason score was ≥ 8 in the 59.6% of cases. There was no treatment discontinuation due to adverse events and all patients demonstrated good tolerance to the drugs. Most patients (86.8%) had a low volume disease with 1 or 2 sites of metastases. All patients were previously treated with ADT for hormone-sensitive prostate cancer, and 156 (70.9%) received docetaxel for CRPC before ARSI. One-hundred ninety (86.4%) men had bone metastasis and 31 (14.1%) visceral metastasis (Table 1). There were no censored patients, at the time of the statistical analysis only 16 patients were alive.

In all pre-treatment plasma samples, median ctDNA fraction assessed by targeted NGS was 0.180 (IQR 0.10–0.39), and *AR* gain was detected in 73 (33.2%) patients.

### 3.2. Plasma DNA Analysis and Clinical Outcomes

In patients with high ctDNA, a lower rate of PSA decline ≥ 50% was detected than patients with low ctDNA (*p* = 0.017). A meaningful shorter PSA doubling time (DT) from nadir was observed in men with high vs. low ctDNA (2.5 vs. 3.8 months, *p* = 0.024), as well as a significant higher PSA DT velocity from nadir (14.3 vs. 3.3 months, *p* = 0.0002). (Table 2) (Appendix A). The higher PSA DT velocity from nadir in patients with high ctDNA versus low ctDNA was maintained both in chemotherapy-naïve and chemotherapy-treated patients.

The time to PSA progression was shorter in patients with with high ctDNA vs. low ctDNA (3.8 vs. 9.1 months, *p* < 0.0001).

Similarly, patients with AR gain experienced a lower rate of PSA decline ≥ 50% compared to patients with AR normal (*p* = 0.0003) and a shorter PSA DT from nadir (2.4 vs. 3.6 months, *p* = 0.001) (Table 3). Both chemotherapy-naïve and chemotherapy-treated patients showed higher PSA DT velocity from nadir in case of AR gain compared to AR normal (12.5 vs. 2.4 months, *p* = 0.045). Furthermore, the time to PSA progression was shorter in patients with AR gain vs. AR normal (3.5 vs. 7.1 months, *p* < 0.0001).

The 3% of patients experienced an initial PSA flare but there is no correlation with clinical outcome.

In the univariate analysis, we highlighted that both AR gain versus AR normal and high ctDNA versus low ctDNA correlates with shorter PFS (3.6 vs. 8.4 months, *p* < 0.0001 and 4.2 vs. 9.5 months, *p* < 0.0001, respectively) (Appendix A).

Combining ctDNA fraction and AR copy number (CN), we identified four distinct groups based on median PSA-PFS: 1) low ctDNA/AR normal, 2) high ctDNA/AR normal, 3) low ctDNA/AR gain, and 4) high ctDNA/AR gain (11.4 vs. 5 vs. 4.8 vs. 3.7 months, *p* < 0.0001). In addition, there were similar results among these four cohorts for radiographic-PFS and OS (27.7 vs. 19.9 vs. 14.9 vs. 8.6 months, respectively *p* < 0.0001) (Figure 3). In a multivariable analysis (Table 4), high ctDNA, AR gain, PSA DT and PSA DT velocity remained independent predictors of PSA-PFS (HR 1.92, 95% CI 1.27–2.9, *p* = 0.002, HR 1.66, 95% CI 1.02–2.7, *p* = 0.04, HR 0.91 96% CI 0.85–0.97, *p* = 0.004, and HR 1.0 95% CI 1.01–1.02, *p* = 0.003, respectively).

## 4. Discussion

One key point of treating CRPC patients is to prolong survival while maintaining the quality of life and preventing needless toxic effects of an ineffective treatment. To reach this aim, the early evaluation of therapeutic efficacy is a crucial goal in the patient management strategy. The current gold standard for assessing tumor response and treatment efficacy is the radiographic imaging and PSA level assessment.

Various retrospective studies have demonstrated that PSA kinetics could be a prognostic biomarker during the history of prostate cancer [40,41,42].

In patients affected by mCRPC and treated with chemotherapy, both PSA velocity and PSA DT provide independent prognostic information [43]. A strong association between PSA kinetics and OS was also demonstrated during treatment with abiraterone in chemotherapy-pretreated and chemotherapy-naïve patients [30].

In the last decades, several efforts are been made to identify new biomarkers with a predictive and/or prognostic role, and the most relevant pathway explored in prostatic cancer depends on AR. An increase in AR copy number, known as AR gain, was detected in up to 70% of CRPC [44,45,46], compared to 1% found in hormone-sensitive prostate cancer [47,48].

The high expression of AR could bypass the inhibition induced by ARSI and finally result in resistance to therapy. PSA may be considered as an excellent biomarker of AR activity [20,49]. Indeed, in our study we revealed a strong association between PSA level and AR CN at baseline reporting higher PSA values in AR-gained compared to AR-normal patients (123.4 ng/mL versus 20.64 ng/mL, *p* < 0.0001). In addition, the present study, including the simultaneous assessment of PSA assessments and ctDNA, offers a significant opportunity to investigate prostate tumor dynamics thanks to a sequential monitoring of PSA during therapy as a surrogate biomarker of the emergence of resistance to treatment. Similar results of a strong correlation between declines in PSA and ctDNA levels have been also observed in a recent study of 140 CRPC patients [50].

Our work showed that changes in PSA dynamics in combination with plasma DNA analysis were independent predictors of outcome and so more reliable early predictive biomarkers in mCRPC, especially in certain conditions.

First, it may be a future valid aid for the subgroup of individuals for whom PSA value does not represent the real condition of the disease; particularly, in aggressive variants of prostate cancer with low or non-rising PSA levels. Second, findings obtained did not show the superiority of one biomarker over the other but rather the combination of ctDNA with PSA kinetics may improve outcome prediction, for example, adding some information about response to treatment in a subgroup of prostate cancer patients without target lesions, such as those with only bone metastatic disease, in which CT scan and bone scan could lack of sensibility and specificity in detection of response or progression. Third, PSA flare is a phenomenon that could deceive clinicians due to an initial and transient rising of PSA levels but with no correlation with poor prognosis [25]. In fact, despite the small number of men harboring PSA surge, in the present study we confirmed ctDNA fraction didn’t mirror clinical phenomena such as PSA or bone flares [51]. Furthermore, we showed no association among PSA level, AR CN and ctDNA fraction in six patients experiencing a PSA flare (five AR-normal and one AR-gained patients, and high ctDNA level in two patients and low in three men; for the patient with AR gain the value of ctDNA was not available).

Lastly, there is growing attention about the possible role of immunotherapy in prostate cancer [51,52]. It is largely demonstrated that the use of these drugs could determine an initial pseudo progression due to inflammation and which does not reflect the real status of the disease. This pattern of response may not be adequately described by traditional response criteria. Therefore, in this condition, monitoring early changes in PSA in association with molecular evidence might be relevant in the management of these patients.

In a previous study, we generated a prognostic score based on ctDNA fraction and functional imaging to better predict treatment outcome [53]. Similarly, we described circulating AR CN, [18F]fluoromethylcholine-PET parameters and other clinical features predicted OS in patients affected by mCRPC and treated with ARSI [54]. However, in the light of this evidence, physicians should also incorporate PSA kinetics within nomograms for predictions of outcome in mCRPC patients receiving abiraterone or enzalutamide and so identify a subgroup of patients who need a close follow-up. This evidence could be applied to the earliest stages of the disease, as demonstrated in a recent post hoc analysis of SPARTAN trial in nonmetastatic CRPC patients receiving apalutamide to assess the relationships between PSA kinetics, outcomes, and molecular classifications using a Decipher genomic classifier and basal/luminal subtypes based on the Decipher test [55,56]. In addition, Antonarakis et al. [57] in real-world individuals with advanced prostate cancer showed that both PSA and ctDNA were additive and independent prognostic factors, prompting a greater benefit from evaluation of both in tandem beyond genomic profiling.

This paper has many limitations: a single-center study, heterogeneous population including chemotherapy-naïve and post-docetaxel patients as well as patients treated with enzalutamide and abiraterone, a limited number of multiple sequential PSA values. Furthermore, there are not longitudinal data concerning ctDNA. However, even these limitations, the present study represents an essential first step towards achieving the introduction of serial ctDNA assessment with standard tests utilized in clinical practice, including PSA dynamics.

## 5. Conclusions

These findings underline the importance of a multimodal approach to outcome prediction by integrating standard imaging and blood tests, such as PSA kinetics parameters, and molecular information to better understand the response to treatment and the efficacy of novel drugs not only in mCRPC but also in early stages of disease. Specifically, the study of liquid biopsy could also serve as a biomarker of disease response to allow for earlier switch or intensification of therapy in patients harboring both plasma AR gain and high ctDNA before starting ARSI.

## Figures and Tables

**Figure 1 cancers-14-02219-f001:**
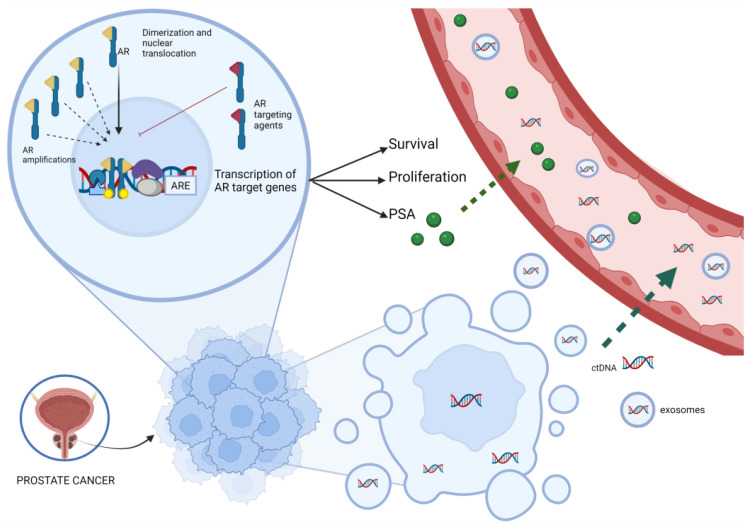
Correlation between ctDNA, AR amplification and PSA kinetics. The picture synthesizes the role of AR and its action mediated by various transcriptional co-regulators. Activation of target genes leads to biological responses including growth, survival and the production of PSA. The AR targeting agents could inhibit this pathway, but the onset of AR amplification can bypass this inhibition. ctDNA is released from dying cancer cells, or, more rarely, from living tumor cells that actively release DNA and exosomes into the circulation because of oncogenic properties. Abbreviations: AR, androgen receptor, ARE, androgen response elements; ctDNA, circulating tumor DNA; PSA, prostate specific antigen.

**Figure 2 cancers-14-02219-f002:**
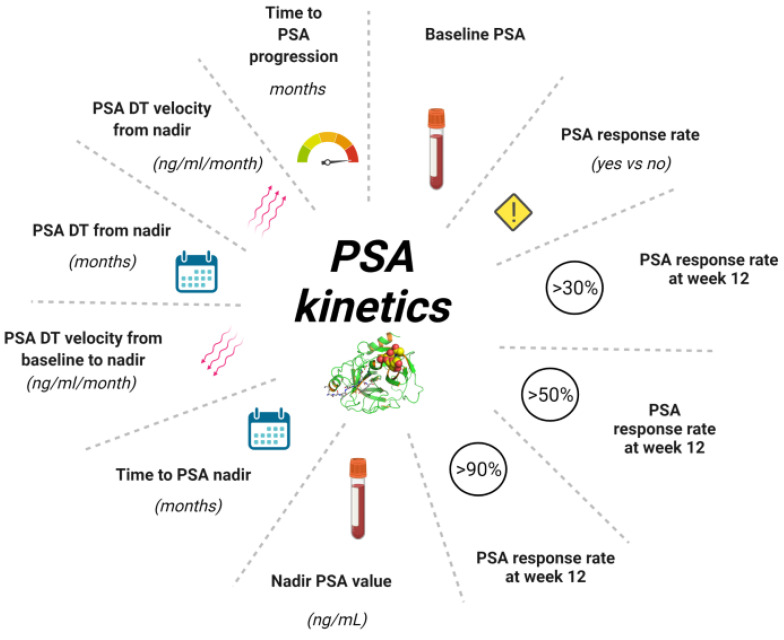
Different parameters of PSA Kinetics. PSA kinetics represents useful tool during the history of prostate cancer from diagnosis to treatment of mCRPC allowing to monitor the follow-up or the efficacy of treatment. Baseline PSA is required at diagnosis and at the beginning of each new treatment. PSA response rate is the eventual decline > 50% in pre-treatment PSA following treatment. PSA response rate at 12 weeks is evaluated in different percentages. The PSA nadir is the lowest value reached during a treatment, another useful indicator is represented by the time in which PSA nadir was reached expressed in months. One of the parameters most commonly used in clinical practice is determination of the dynamics of PSA levels, expressed by PSADT and PSA velocity. These are routinely used in early setting but could be useful also in advanced setting. Abbreviation: PSA, prostate specific antigen; PSA DT, PSA doubling time; mCRPC, metastatic castration resistant prostate cancer.

**Figure 3 cancers-14-02219-f003:**
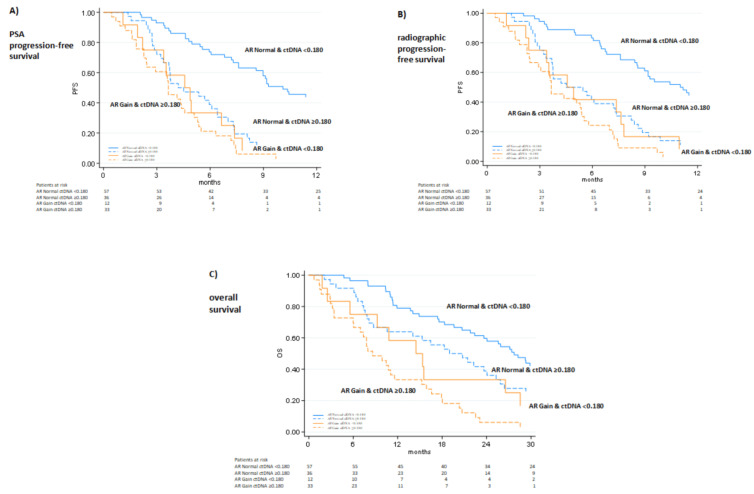
Combining ctDNA fraction/AR status and PSA kinetics (**A**), radiographic progression-free survival (**B**) and overall survival (**C**). Radiographic and PSA response was assessed according to PCWG3 criteria.

**Table 1 cancers-14-02219-t001:** Patient characteristics.

	N (%)
Age, years	
<74 *	104 (47.3)
≥74	116 (52.7)
Prostatectomy	
No	119 (55.1)
Yes	97 (44.9)
Unknown/missing	4
Radiotherapy	
No	162 (75.0)
Yes	54 (25.0)
Unknown/missing	4
Gleason score	
6–7	80 (40.4)
≥8	118 (59.6)
Unknown/missing	22
ECOG PS	
0–1	205 (93.2)
2	15 (6.8)
Sites of metastasis	
Bone	190 (86.4)
Lymph nodes	129 (58.6)
Visceral	31 (14.1)
Liver	17 (7.7)
Lung	14 (6.4)
Other	6 (2.7)
Number of sites of metastasis	
1	101 (45.9)
2	91 (41.3)
3	11 (5.0)
4	12 (5.5)
5	5 (2.3)
Prior lines of therapy	
0	60 (27.3)
1	71 (32.3)
2	38 (17.2)
3	41 (18.6)
4	5 (2.3)
5	5 (2.3)
Prior docetaxel	
No	64 (29.1)
Yes	156 (70.9)
PSA, median value (IQR), ng/dL	36.18 (11.3–154.5)
ALP, U/L	
<129 ^#^	127 (58.8)
≥129	89 (41.2)
Unknown/missing	4
Albumin, g/dL	
≥4	109 (50.2)
<4	108 (49.8)
Unknown/missing	3
Hemoglobin, g/dL	
≥12.5 ^#^	67 (32.8)
<12.5	137 (67.2)
Unknown/missing	16
Serum CgA, ng/mL	
<120 ^#^	100 (46.5)
≥120	115 (53.5)
Unknown/missing	5
LDH, U/L	
<225 ^#^	156 (71.6)
≥225	62 (28.4)
Unknown/missing	2
NLR	
<3	115 (52.3)
≥3	105 (47.7)
ctDNA	
<0.180 *	69 (50.0)
≥0.180	69 (50.0)
Unknown/missing	82
AR CN	
Normal	147 (66.8)
Gain	73 (33.2)

* Median value. ^#^ Upper normal value. Abbreviations. ALP, alkaline phosphatase; *AR*, androgen receptor; CgA, chromogranin A; ECOG, Eastern Cooperative Oncology Group; IQR, interquartile range; LDH, lactate dehydrogenase; N, number; NLR, neutrophil-lymphocyte ratio; PS, performance status; PSA, prostate-specific antigen; ctDNA, circulating tumor DNA.

**Table 2 cancers-14-02219-t002:** Correlation of baseline ctDNA fraction and PSA kinetics.

	ctDNA Low (<0.180)	ctDNA High (≥0.180)	*p*
Baseline PSA (ng/mL) (IQR)	20.64 (9.25 to 95.0)	51.79 (13.79 to 158.0)	0.011
PSA response rate (yes vs. no), n (%)	43 (63.2) vs. 25 (36.8)	29 (42.7) vs. 39 (57.3)	0.017
Maximum % PSA decline (IQR)	−80.92 (−92.74 to −54.48)	−63.11 (−98.23 to −31.94)	0.134
PSA response rate at week 12 (>30%), n (%)	45 (83.3) vs. 9 (16.7)	34 (75.6) vs. 11 (24.4)	0.340
PSA response rate at week 12 (>50%), n (%)	41 (75.9) vs. 13 (24.1)	28 (62.2) vs. 17 (37.8)	0.142
PSA response rate at week 12 (>90%), n (%)	18 (33.3) vs. 36 (66.7)	8 (17.8) vs. 37 (82.2)	0.081
Nadir PSA value (ng/mL) (IQR)	3.62 (1.23 to 17.00)	21.77 (2.69 to 57.32)	0.0008
Time to PSA nadir (months) (IQR)	3.68 (1.81 to 7.34)	1.84 (0.92 to 3.39)	0.012
PSA nadir DT (months) (IQR)	−2.26 (−3.81 to −1.00)	−1.39 (−3.15 to −0.77)	0.007
PSA DT velocity from baseline to nadir (ng/mL/month) (IQR)	−2.90 (−1.46 to −11.30)	−11.53 (−2.89 to −2.28)	0.034
PSA DT from nadir (months) (IQR)	3.80 (2.20–6.20)	2.50 (1.60–3.80)	0.024
**PSA DT velocity from nadir (ng/mL/month) (IQR)**	**3.30 (0.9–13.8)**	**14.3 (2.7–81.4)**	**0.0002**
**Time to PSA PD (months) (IQR)**	**9.11 (4.87 to 16.10)**	**3.78 (2.76 to 7.00)**	**<0.0001**

Abbreviations. DT, double timing; IQR, interquartile range; PD, progression disease; PSA, prostate specific antigen; ctDNA, circulating tumour DNA.

**Table 3 cancers-14-02219-t003:** Correlation of plasma *AR* copy number and PSA kinetics.

	*AR* Normal	*AR* Gain	*p*
Baseline PSA (ng/mL) (IQR)	20.64 (7.16 to 77.00)	123.40 (34.69 to 291.0)	<0.0001
PSA response rate (yes vs. no), n (%)	79 (54.5) vs. 66 (45.5)	21 (28.8) vs. 52 (71.2)	0.0003
Maximum % PSA decline (IQR)	−78.93 (−91.26 to −45.16)	−52.36 (−74.58 to −22.99)	0.003
PSA response rate at week 12 (>30%), n (%)	91 (81.2) vs. 21 (18.8)	26 (65.0) vs. 14 (35.0)	0.037
PSA response rate at week 12 (>50%), n (%)	77 (68.7) vs. 35 (31.3)	20 (50.0 vs. 20 (50.0)	0.035
PSA response rate at week 12 (>90%), n (%)	35 (31.2) vs. 77 (68.8)	4 (10.0) vs. 36 (90.0)	0.008
Nadir PSA value (ng/mL) (IQR)	3.04 (0.95 to 19.53)	47.09 (14.85 to 149.30)	<0.0001
Time to PSA nadir (months) (IQR)	2.96 (1.76 to 6.61)	1.81 (0.92 to 2.55)	0.0002
PSA nadir DT (months) (IQR)	−2.19 (−3.91 to −1.00)	−1.44 (−3.44 to −0.77)	0.463
PSA DT velocity from baseline to nadir (ng/mL/month) (IQR)	−3.02 (−1.43 to −1.26)	−16.68 (−6.47 to −4.06)	0.001
PSA DT from nadir (months) (IQR)	3.60 (2.10–7.71)	2.40 (1.40–3.40)	0.001
**PSA DT velocity from nadir (ng/mL/month) (IQR)**	**2.8 (0.7–14.3)**	**27.8 (9.6–116.1)**	**<0.0001**
**Time to PSA PD (months) (IQR)**	**7.07 (3.78 to 13.09)**	**3.55 (1.91 to 6.22)**	**<0.0001**

Abbreviations. *AR*, androgen receptor; DT, double timing; IQR, interquartile range; PD, progression disease; PSA, prostate specific antigen.

**Table 4 cancers-14-02219-t004:** Multivariate analysis of PSA progression-free survival.

	PFS		OS	
	HR (95% CI)	*p*	HR (95% CI)	*p*
**Age** (≥74 vs. <74)	0.98 (0.96–1.01)	0.218	1.01 (0.98–1.04)	0.394
**logPSA**	1.07 (0.95–1.21)	0.254	1.06 (0.94–1.20)	0.342
**Visceral metastasis** (yes vs. no)	1.22 (0.71–2.09)	0.476	1.85 (1.06–3.23)	0.029
**Previous chemotherapy** (yes vs. no)	1.29 (0.85–1.96)	0.226	1.14 (0.74–1.76)	0.559
**ECOG PS** (2 vs. 0–1)	1.33 (0.92–1.93)	0.128	1.58 (1.08–2.32)	0.018
**ALP** (≥129 vs. <129)	0.98 (0.64–1.51)	0.933	1.02 (0.67–1.56)	0.933
***AR*****CN** (Gain vs. Normal)	1.72 (1.05–2.81)	0.031	1.44 (0.86–2.40)	0.162
**ctDNA** (>0.180 vs. ≤0.180)	4.64 (1.53–14.06)	0.007	3.50 (1.14–10.77)	0.029
**LDH** (≥225 vs. <225)	2.13 (1.24–3.64)	0.006	1.94 (1.16–3.25)	0.012
**PSA DT from nadir** (continuous variable)	0.93 (0.87–0.98)	0.015	0.93 (0.87–0.99)	0.030
**PSA DT velocity from nadir** (continuous variable)	1.001 (1.000–1.002)	0.066	1.000 (0.999–1.001)	0.572

Abbreviations. AR CN, androgen receptor copy number; CI, confidence interval; DT, doubling time; HR, hazard ratio; LDH, lactate dehydrogenase; PFS, progression-free survival; PSA, prostate-specific antigen; ctDNA, circulating tumour DNA; UNL, upper normal limit.

## Data Availability

The anonymized datasets used and/or analyzed during the current study are available from the corresponding author upon reasonable request.

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
