# Peer review of "Baseline Plasma Tumor DNA (ctDNA) Correlates with PSA Kinetics in Metastatic Castration-Resistant Prostate Cancer (mCRPC) Treated with Abiraterone or Enzalutamide"

_cancers, 2022, doi:10.3390/cancers14092219_

Round 1
Reviewer 1 Report
Conteduca et al. presented were interesting study analyzing the use of ctDNA and PSA, including its kinetics, as predictive factors in patients with prostate cancer. The topic is very important and will be of great interest to readers. AUthors collected a large group of over 200 patients and collected a wide set of longitudinal data. I admire the effort put in conducting the study, however, I have multiple comments that should be addressed before consideration of the manuscript for publication in Cancers.
Major:
- The most important issue (next points are reported in the order as found in the manuscript): Why baseline PSA level was not included in the regression analysis? Observing high differences in baseline PSA levels between ctDNA low and high or AR normal vs gain groups, and biological bases of ctDNA and PSA secretion, I would anticipate that both biomarkers highly correlate with each other. There is a possibility that magnitude of effect of baseline ctDNA level would be much lower or even lost if PSA was included in the model. Please include it in the model, provide both models and explain why it was not included.
- Page 1, line 40 “Prostate cancer is one of the most malignant tumor in men and the second leading cause of cancer-related death worldwide” – I’m not aware of classification of more or less malignant tumor. Do you mean most common malignant tumor?
- Two different abbreviations are used for circulating tumor DNA in plasma (ctDNA and ptDNA; e.g. fig. 1 has ctDNA in the picture but ptDNA in the description). Please select one and use consistently. ctDNA is more commonly used in the literature thus seems more appropriate and can increase impact of the study (and citations)
- Study requires extensive language editing. Introduction should be re written since it’s hard to follow. Some sentences seems to have not any connection with previous or next ones. This could be caused by translation from other language thus revision by native speaker could help to improve clarity.
- Was “start” PSA evaluated before the start of treatment or within first 3 days of treatment? It’s not clear from current description. Was plasma for ctDNA collected at the same time as for PSA?
- Description of results in point 3.2 is completely unclear and hard to read. I would recommend following the scheme from Tables 2 and 3. First describe data for ctDNA low vs high, than for AR gain or AR normal. There is no need to repeat all data from tables with numbers. Better to focus on most important findings and refer to table for other.
- I don’t see a point of presenting a case to illustrate results of the study. If authors intend to do it, I would recommend to move it to supplementary materials since now it’s distracting the reader. For study with over 200 cases, presenting a single case an example is not a standard and those not increase its quality, rather distract reader.
- Please mark statistically significant results in table 2 and 3
- Figure 3 is missing marks for censored patients
- How authors selected variables for multivariate COX regression? Was univariate analyses performed before? Please describe the selection of variables in the methods. If univariate analyses was performed, please report results (could be done also in the supplementary materials)
- Can authors provide also multivariate analyses for OS? It would be valuable to see if predictive groups reported here are also prognostic.
- All reported analyses are based on pre-treatment ctDNA but in methods authors stated that serial samples were collected. Are there any longitudinal data available? Were there any correlations between decrease in PSA level and decrease in ctDNA? Was the time to PSA nadir correlated to time to ctDNA nadir?
- Conclusions reported in lines 261-266 re not justified by results. Authors did not showed that ctDNA is better than PSA, neither any data about better response evaluation in patients without targeted lesions (subgroup analyses would be required)
- Authors discuses PSA flair but not presented any data on this topic in the results. Please provide results or remove it from discussion
- Discussion is missing information about other studies assessing the same issue (in same or different settings)
Minor:
- Each abbreviations should be described both in abstract and in main text (describing it in the abstract only is not enough). Full names should be used in keywards, not abbreviations, when possible
- Please include in methods the information about censoring
- Line 169 should be metastases, not metastasis
Reviewer 2 Report
The manuscript by Conteduca et.al. showed that baseline plasma tumor DNA and androgen receptor copy number were associated with worse outcomes in a prospective mCRPC patient cohort at a single institution. To do so, they explored the association between these two parameters with PSA kinetics and survival data.
Major comments
- Did all patients tolerate abi or enza treatment? Any dropouts?
- The authors included "prior lines of therapy" in Table 1. Could they list what exact therapies patients received?
- There was no description of how AR CN was analyzed in methods.
- In discussion, the authors mentioned the utility of ptDNA and AR CN in "prostate cancer with low or non-rising PSA". The patient cohort described in the study had a median PSA of 36 so it is difficult extrapolate the current finding to a low PSA scenario.
- The take home message of this paper is that ptDNA and AR CN is associated with worse outcome. Could the authors comment on how this additional information could potentially change current clinical practice? In other words, would a patient with high ptDNA and AR CN gain be treated differently than a patient with low ptDNA and normal AR CN?
Minor comments
- Were the PFS reported in line 33 median survival time? If so, please specify.
- Figure 1, ctDNA was used instead of ptDNA.
Round 2
Reviewer 1 Report
Thank you for considering my previous comments. Article is well fitted for publication